# Sugarcane Straw Polyphenols as Potential Food and Nutraceutical Ingredient

**DOI:** 10.3390/foods11244025

**Published:** 2022-12-13

**Authors:** Ana L. S. Oliveira, Maria João Carvalho, Diana Luazi Oliveira, Eduardo Costa, Manuela Pintado, Ana Raquel Madureira

**Affiliations:** CBQF—Centro de Biotecnologia e Química Fina—Laboratório Associado, Universidade Católica Portuguesa, Escola Superior de Biotecnologia, Rua Diogo de Botelho 1327, 4169-005 Porto, Portugal

**Keywords:** *Saccharum officinarum* L., straw, phenolic compounds, antidiabetic, antityrosinase, anti-inflammatory

## Abstract

The sugarcane processing industry generates a large amount of straw, which has a negative environmental impact, and high costs are associated with their elimination, wasting their potential bioactive value attributed to their richness in polyphenols. In this study, an ethanolic extract produced from sugarcane straw was screened for its phenolic compounds content, and the potential use of this extract in the development of a food ingredient was further evaluated. Fifty different secondary metabolites belonging to the hydroxybenzoic acids, hydroxycinnamic acids, and flavonoids were identified by liquid chromatography–electrospray ionization–ultrahigh-resolution—quadrupole time of flight–mass spectrometry (LC-ESI-UHR-QqTOF-MS). The predominant phenolic compounds found were 4-hydroxybenzaldehyde, chlorogenic acid, and 5-*O*-feruloylquinic acid. The obtained extracts showed strong potential as food preservatives by exhibiting (a) antioxidant activity using both 2.2′-azino-bis (3-ethylbenzothiazoline-6-sulphonic acid) diammonium salt radical cation (ABTS) and 2,2-diphenyl-1-picrylhydrazyl (DPPH) methods; and (b) antimicrobial capacity, with a minimum inhibitory concentration of 50 mg/mL for *Staphylococcus aureus*, 74% inhibition for *Bacillus cereus*, and 44% for *Salmonella enterica*; and (c) the capacity to inhibit a food browning enzyme, tyrosinase (28–73% for 1–8 mg/ mL). Moreover, the extracts showed antidiabetic potential by inhibiting the enzymes α-glucosidase (15–38% for 1.25–5.00 mg/mL) and dipeptidyl peptidase-IV (DPP-IV) (62–114% for 0.31–5.00 mg/mL). The extract (0.625 mg/mL) also exhibited the capacity to reduce proinflammatory mediators (i.e., interleukins 6 and 8, and tumor necrosis factor alpha) when Caco-2 cells were stimulated with interleukin 1 beta. Thus, sugarcane straw extract, which is rich in phenolic compounds, showed high potential to be used in the development of food-preservative ingredients owing to its antioxidant and antimicrobial potential, and to be explored as a food supplement in diabetes prevention and as coadjuvant to reduce intestinal inflammation by reducing proinflammatory mediators.

## 1. Introduction

Sugarcane (*Saccharum officinarum* L.) is one of the most widely cultivated crops and represents 70% of the global sugar market [1]. Sugar production generates significant quantities of different byproducts, including straw residues, which are usually discarded after harvesting. However, the phytochemical profile of straw comprises a myriad of value-added bioactive compounds including, but not limited to, phenolic compounds [2], making straw residues a rather cheap source of functional molecules. The phenolic compounds can be isolated, and their potential exploited for further food systems applications. The phenolic compounds profiles of several sugarcane byproducts have been characterized, mainly those of leaves, bagasse, and molasses, revealing the presence of flavones (luteolin, apigenin, and tricin derivatives), phenolic acids (sinapic, caffeic, and ferulic acids), and hydroxycinnamic acids [3,4]. However, sugarcane straw polyphenolic richness has not been well exploited, most likely because the straw residues were left in the fields to improve soil quality, increasing organic matter, water storage, control of weed infestation and, soil erosion, while reducing the amount of potassium and nitrogen fertilizers required. However, this practice also presents disadvantages, such as attracting pests and increasing N_2_O emissions [5].

The phenolic compounds remaining in straw residues can be recovered and applied in a variety of food systems, increasing sugarcane’s straw value. Food manufacturers use food-grade antioxidants to prevent lipid components and quality deterioration while maintaining nutritional value. There is always a desire to replace synthetic compounds (e.g., butylated hydroxyanisole (BHA), butylated hydroxytoluene (BHT), propyl gallate (PG), and tert-butylhydroquinone (TBHQ)) by those that naturally inhibit oxidation [6].

One of the main challenges in the food industry is the enzymatic browning that occurs in fruits and vegetables [7,8]. Antioxidants and browning inhibitors are widely employed in the food industry to inactivate the enzymes responsible for browning (e.g., PPO type, tyrosinase), especially in the fresh-cut fruit industry [9,10]. Therefore, straw polyphenols may have additional technological applications in the food industry. In fact, one way to improve the shelf life and safety of fruits and vegetables with natural ant-browning/antioxidant properties is trough dipping. This process may also be a way to treat microbial contamination and simultaneously provide an alternative to synthetic food additives [7].

Another target for natural extracts in food markets is food supplements, which are used in the prevention and/or amelioration of certain health problems or to enrich food’s nutritional value. Diabetes, which is a worldwide chronic disease, affected 463 million people in 2019, and this number is expected to reach 700 million by 2045 [1]. Type 2 diabetes is the most prevalent form, accounting for approximately 90% of cases [11]. An estimated USD 850 billion was spent globally on the treatment and health interventions for diabetes in 2017 [12]. Thus, natural-food-derived molecules such as flavonoids, alkaloids, terpenoids, anthocyanins, glycosides, and phenolic compounds, with capacity to inhibit enzymes responsible for carbohydrate catalyzes into glucose (α-glucosidase or dipeptidyl peptidase-IV (DPP-IV)) show superiority over some clinical drugs due to their safety and reduced risks of side effects [13]. Inhibitors of both α-glucosidase and DPP-IV may have more consistent efficacy to reduce postprandial hyperglycemia, independent of the types of carbohydrate contained in a meal [14].

Dietary factors may contribute to the development of many chronic diseases associated with local and systemic inflammation that can be mediated by reactive oxygen species (ROS). Phenolic compounds not only directly scavenge free radicals but also regulate cell redox balance and synthesis of inflammation-related cytokines [15].

Over the last few decades, a wide variety of functional foods, supplements, and nutraceuticals has emerged in the global market, and this segment is expected to reach about USD 210 billion in 2026 [16], with Asia and North America being the main food market consumers in the world. Nowadays, Europe is focused on gathering clinical evidence regarding the health benefits and safety of emerging functional products [17], as the food industry seeks high stability, functionality, and user-friendly food additives, preferably from natural sources, with a wide range of applications [15]. In this work, our aim was to identify the phenolic compound profile of sugarcane straw ethanolic extracts. The extracts were characterized according to their potential techno-functionally as food preservatives, or bio-functionality as a nutritional supplement for health promotion. Thus, the experimental study included the following tasks: (a) identification of individual polyphenols and quantification of the most relevant compounds (LC-ESI-UHR-QqTOF-MS); (b) determination of antioxidant capacity (ABTS and DPPH); (c) determination of antimicrobial activity against *Salmonella enterica serovar Enteritidis, Escherichia coli*, *Staphylococcus aureus, Bacillus cereus* and *Pseudomonas aeruginosa*; (d) determination of antibrowning capacity with an in vitro tyrosinase inhibitory activity; (e) determination of antidiabetic potential through the inhibition of α-glucosidase and DPP-IV; and (f) determination of the downregulation of inflammatory cytokines (interleukins 6 (IL-6) and 8 (IL-8), and tumor necrosis factor alpha (TNF-α)) in Caco-2 cells.

## 2. Materials and Methods

### 2.1. Chemicals and Reagents

The 2-azinobis-3-ethylbenzothiazoline-6-sulphonic acid (ABTS), aluminum trichloride, ascorbic acid, 2,2-diphenyl-1-picrylhydrazyl (DPPH), food-grade ethanol (99%), gallic acid, α-glucosidase from *S. cerevisiae, p*-coumaric acid, trifluoroacetic acid, quercetin, sodium carbonate (Na_2_CO_3_), and Trolox were purchased from Sigma-Aldrich (Sintra, Portugal). Acetonitrile was purchased from Fischer Scientific (Oeiras, Portugal). Folin-Ciocalteu reagent and potassium persulfate (K_2_S_2_O_8_) were purchased from Merck (Algés, Portugal).

The standards of vitexin, diosmetin, isoschaftoside, orientin, viexin-2-*O*-rhamnoside (Extrasynthése, Genay, France), tricin, luteolin, protocatechuic acid, vanillic acid, *p*-coumaric acid, caffeic acid, ferulic acid, chlorogenic acid, gentisic acid, 4-hydroxybenzaldehyde, 4-hydroxybenzoic acid, and 3,4-dihydroxybenzalhedyde, syringic acid (Sigma-Aldrich) were used as external standards for calibration curves in LC-ESI-UHR-QqTOF-MS.

### 2.2. Byproduct Material

Sugarcane (*Saccharum officinarum* L.) straw was provided by Raízen from Guariba, São Paulo, Brazil. Samples were collected in October 2020.

Upon arrival to the CBQF-UCP laboratory (Porto, Portugal), the material was dried at 40 °C for 12 h using a ventilated oven (Memmert GmbH + Co., KG, Schwabach, Germany), which was followed by a milling process using a grinder (SM100, Retsch, Vila Nova de Gaia, Portugal) to obtain a particle size < 4 mm. Straw was stored at room temperature and protected from light until the beginning of assays.

### 2.3. Extraction and Isolation of Phenolic Compounds from Sugarcane Straw

In brief, dried sugarcane straw powder was extracted with 50% (*v*:*v*) ethanol in a ratio biomass:solvent of 1:10 (*w*:*v*) during 24 h at 30 °C under agitation at 120 rpm (Innova 40 New Brunswick, Eppendorf, Hamburg, Germany) and protected from light. The extraction process was repeated three times. The solid and liquid fractions were separated by filtration with gauze, and the liquid fraction was centrifuged at 18.671× *g* for 10 min (Sorvall Lynx 4000 centrifuge, Thermo Scientific, Waltham, MA, USA). The ethanol was removed from the liquid fraction evaporation under vacuum with a rotary evaporator at 50 °C and 150 mbar (Heidolph, Walpersdorfer, Germany).

The obtained aqueous fraction was further applied to an amberlite XAD-2 (Sigma-Aldrich, St. Louis, MI, USA) resin for the next purification. The amberlite XAD-2 was washed with methanol followed by three washes with deionized water. Resin was preconditioned 12 h in ultra-pure water before being used. The resin was used in a ratio of 1:2 (*v:w*) and left under agitation of 100 rpm overnight at room temperature. After that, it was isolated and washed twice with deionized water at pH 2 to remove any adsorbed sugar. The desorption of the phenolic compounds was performed in two steps: a first with 50% (*v*:*v*) ethanolic solution acidified at pH 2 (HCL, 10M) under agitation at 100 rpm, 37 °C, overnight; a second desorption with 50% (*v*:*v*) ethanolic solution. The ethanolic extracts were recovered by decantation and filtration (type I filter, V Reis, Lisbon, Portugal) and combined. The ethanol was evaporated with a rotary evaporator (50 °C; 150 mbar), and the dried extracts were obtained by freeze drying (Martin Christ, Osterode am Harz, Germany) for further characterization.

### 2.4. Phenolic Compounds and Organic Acid Identification and Quantification by LC-ESI-UHR-QqTOF-MS

The identification and quantification of all phenolic compounds were performed by liquid chromatography–electrospray ionization–ultrahigh-resolution–quadrupole time of flight–mass spectrometry (LC-ESI-UHR-QqTOF-MS) [18]. For this, the dried extracts were previously dissolved in a solution of 50% (*v*:*v*) ethanol at a final concentration of 50 mg/mL and filtered through 0.45 µm filters before injection

The separation was performed with a Bruker Elute series equipped with a UHR-QqTOF mass spectrometer (Impact II, Bruker Daltonics, Bremen, Germany) and a BRHSC18022100 intensity Solo 2 C18 column (100 × 2.1 mm, 2.2 μm, Bruker). Separation was carried out at a flow rate of 0.25 mL/min with the following elution gradient: 0 min, 0% B; 10 min, 21.0% B; 14 min, 27% B; 18.30 min, 58%; 20.0 min, 100%; 24.0 min, 100%; 24.10 min, 0%; 26.0 min, 0% and mobile phase A (0.1% (*v*:*v*) aqueous formic acid) and B (acetonitrile with 0.1% (*v*:*v*) formic acid). The MS acquisition was set to negative ionization mode with the selected parameters: end plate off set voltage, 500 V; capillary voltage, 3.0 kV; drying gas temperature, 200 °C; drying gas flow, 8.0 L/min; nebulizing gas pressure, 2 bar; collision radio frequency (RF), from 250 to 1000 Vpp; transfer time, from 25 to 70 μs; collision cell energy, 5 eV. For the internal mass calibration, we used sodium formate clusters.

The elemental composition was confirmed according to accurate mass and isotope rate calculations designated mSigma (Bruker Daltonics), and phenolic compounds were identified based on their accurate mass [M−H]^−^; the results were expressed in micrograms per gram of dry extract. The standards used for external calibration were vitexin, diosmetin, isoschaftoside, orientin, viexin-2-*O*-rhamnoside, tricin, luteolin, protocatechuic acid, vanillic acid, *p*-coumaric acid, caffeic acid, ferulic acid, chlorogenic acid, 2,5-dihydroxybenzoic acid, 4-hydroxybenzaldehyde, 4-hydroxybenzoic acid, 3,4-dihydroxybenzalhedyde, and syringic acid.

### 2.5. Antioxidant Capacity Evaluation

#### 2.5.1. ABTS Radical Cation Decolorization Assay

The 2,2′-azino-bis (3-ethylbenzothiazoline-6-sulphonic acid) diammonium salt radical cation (ABTS) decolorization assay was performed through the generation of free radicals after 16 h of reaction in dark between a solution of ABTS (7 mmol/L) and a solution of potassium persulfate (2.45 mM) in a 1:1 (*v/v*) proportion. The diluted ABTS (OD of 0.700 ± 0.020 at 734 nm) was reacted with samples in a 96-well microplate (15 μL of each sample in duplicate reacted with 200 μL of ABTS) and incubated for 5 min at 30 °C, and measurements were taken in a microplate reader (Synergy H1 microplate reader, Biotek, Winooski, Vermont, USA) at 734 nm [19].

#### 2.5.2. DPPH Radical Cation Decolorization Assay

The 2,2-diphenyl-1-picrylhydrazyl (DPPH) radical cation decolorization assay was performed with a solution of DPPH (600 μM) in methanol, and then was diluted to a final OD of 0.600 ± 0.100 at 515 nm. Then, in a 96-well microplate, each sample (25 μL) was mixed with DPPH (175 μL) in duplicate. After a 30 min incubation period at 25 °C, the OD was measured at 515 nm with a microplate reader (Synergy H1 microplate reader, Biotek, Winooski, VT, USA) [19].

For each antioxidant method, the inhibition percentage of the free radicals was calculated according to equation (1) for each sample, and a calibration curve was prepared using Trolox standard solutions (0.075–0.008 mg/mL). The results are expressed as milligrams of Trolox equivalent per gram of dry extract (mg TE/g dry extract). Two independent analyses were performed in each of the triplicate extracts obtained.
(1)$$I\left(\%\right)=\left[\frac{\left(Abs_{A0}-Abs_{sample}\right)}{Abs_{A0}}\right]\times 100$$
where *Abs_A_*_0_ is the absorbance of blank, and *Abs_sample_* is the absorbance of the reaction between the sample and the radicals.

### 2.6. Minimal Inhibitory and Bactericidal Concentrations Determination

The microorganisms tested in this study were five typical food pathogen and contaminant bacteria: *Salmonella enterica* serovar Enteritidis ATCC 13076, *Escherichia coli* ATCC25922, *Staphylococcus aureus* ATCC 25923, *Bacillus cereus* ATCC14579, and *Pseudomonas aeruginosa* ATCC10145, from the CBQF-ESB collection.

The minimum inhibitory concentrations (MICs) were determined using a broth microdilution assay, following the standards for antimicrobial susceptibility testing provided by the Clinical and Laboratory Standards Institute (CLSL) [20]. For this analysis, the extract powders that were richer in polyphenols were r-dissolved in Muller-Hilton medium (CAMHB-cation-adjusted MHB) for the concentrations ranging from 10 to 50 mg mL^−1^ and sterilized by filtration through a 0.22 µm filter (FriLabo—Maia, Portugal). Each well of a microplate was filled with a total volume of 200 µL containing approximately 5 × 10^5^ CFU mL^−1^ of test bacteria and variable concentrations of the extract prepared in CAMHB. The microplate was incubated for 24 h at 37 °C. The MIC value corresponded to the lowest extract concentration that inhibited visible bacterial growth. Bacterial cells viability was read every hour in a UV/VIS microplate reader (Synergy H1, Biotek, Vila Nova de Gaia, Portugal) with an optical density of 660 nm. Two sterility controls were prepared: one with the medium and another with the extract. The medium with the inoculum was the negative control. All assays were performed in duplicate.

### 2.7. Inhibition of Tyrosinase Activity Quantification

The assay was performed with a commercial tyrosinase inhibitor screening kit (colorimetric) (ab204715, abcam). Briefly, a mixture of dry extract, previously dissolved and diluted in the kit buffer (20 μL) and tyrosinase solution (50 μL), was incubated in a 96-well microplate at 25 °C for 10 min. Then, 30 μL of substrate solution was added and the absorbance was continuously read at A510 nm in a microplate reader (Synergy H1 microplate reader, Biotek, Winooski, Vermont, USA). Data were recorded in 2 min intervals for 30 min. Inhibition control (IC) was performed with Kojic acid (20 μL, 0.021 mg/ mL). Enzyme control was performed with mixture of enzyme and substrate in the presence of buffer.

The average was used for reading duplicates. Two time points (T1 and T2) were chosen in the linear range of the plot, and we obtained the corresponding values for absorbance (A1 and A2).

The slope was calculated for all samples (S), inhibition control (IC), and enzyme control (EC) by dividing the net ΔA (A2–A1) values by the time ΔT (T2–T1). The percentage inhibition of these assays was calculated by Equation (2):(2)$$Enzyme\;inhibition\;activity\;\left(\%\right)=\frac{Slope\;of\;EC-Slope\;of\;S}{Slope\;of\;EC}\times 100$$

### 2.8. Antidiabetic Activity Quantification

The dry extract was previously dissolved and diluted in the buffer corresponding to each test, and the final concentrations tested were between 0.31 and 5 mg/mL All analysis were performed in triplicate.

#### 2.8.1. α-Glucosidase Inhibition Assay

The *α*-glucosidase inhibition was calculated using a colorimetric-based quantitative method with α-glucosidase from *S. cerevisiae* [21]. A total of 50 µL of sample solution and 100 µL of 0.1 M phosphate buffer (pH 6.9) containing α-glucosidase solution (1.0 U/mL) were incubated in 96-well plates at 25 °C for 10 min. After preincubation, 50 µL of a 5 mM ρ-nitrophenyl-α-D-glucopyranoside solution prepared in 0.1 M phosphate buffer (pH 6.9) was added to each well. The plate was incubated at 25 °C for 5 min, and absorbance readings were recorded at 405 nm with a microplate reader (Synergy H1 microplate reader, Biotek, Winooski, VT, USA). Results were compared with the control, which contained 50 µL of buffer solution instead of an active extract. Acarbose was used as the positive standard in a concentration range between 0.078 and 2.500 mg/mL. The α-glucosidase inhibitory activity is expressed as percentage of inhibition, using Equation (3):(3)$$Inhibition\;\left(\%\right)=\frac{\Delta Abs_{CNT}-\Delta {\mathrm{Abs}}_{S}}{\Delta Abs_{CNT}}\times 100$$

#### 2.8.2. Dipeptidyl Peptidase-IV (DPP-IV) Inhibition Assay

The assay was performed with a commercial DPP (IV) inhibitor screening assay kit (Cat. 700210, Cayman Chemical, Tallinn, Estonia). Briefly, 10 μL aliquots of different concentrations of the active extracts were mixed with 10 μL of DPP-IV and 30 μL of assay buffer. After the addition of 50 μL of substrate solution (100 mM of Gly-Pro-p-nitroanilide in Tris-HCl buffer (pH = 8.0)) to initiate the reaction, the plate was incubated for 30 min at 37 °C, and the fluorescence was read using an excitation wavelength of 350–360 nm and an emission wavelength of 450–465 nm (Synergy H1 microplate reader, Biotek, Winooski, VT, USA). Sitagliptin was used as the positive control (10–100 μM).

The enzyme and sample solution of the sample control group and the enzyme solution of the blank control group were replaced with buffer, whereas all the remaining reagents were kept equal to those of the sample group.

The background fluorescence was subtracted to the 100% initial activity for each inhibitor well. The inhibition rates of the enzymes were calculated according to Equation (4).
(4)$$Inhibition\;\left(\%\right)=\left[\frac{initial\;activity-inhibitor}{initial\;activity}\right]\times 100$$

### 2.9. Cytotoxicity Evaluation

Cytotoxicity was evaluated according to the International Organization for Standardization (ISO, 2009), as previously described for Caco-2, HT29-MTX, and Hep G2 cells [22,23]. Cells were grown to 80–90% confluence, detached using TrypLE Express (Thermo Scientific, Massachusetts, USA), and seeded at 1 × 10^4^ cells/well in a 96-well microplate (Nunclon Delta, Thermo Scientific, Waltham, MA, USA). After 24 h, the culture media were carefully removed and replaced with culture media supplemented with CMC at concentrations between 1.56 and 50 mg/mL. DMSO (Sigma, St. Louis, MI, USA) at 10% (*v*:*v*) in culture media was used as a death control, and plain culture medium was used as the growth control. After 24 h of incubation, 10 µL of Presto Blue (Thermofisher, Waltham, MA, USA) was added to each well and incubated. After this period, fluorescence (excitation: 560 nm; emission: 590 nm) was measured using a microplate reader (Synergy H1 microplate reader, Biotek, Winooski, VT, USA). All assays were performed in quadruplicate.

### 2.10. Caco-2 Monolayer Immunomodulation

The immunomodulatory assays were performed with Caco-2 cells that were seeded at 2.5 × 10^5^ cells/well in a 24-well microplate and incubated for 24 h at 37 °C [24]. Following this, the culture media were carefully replaced with media supplemented with extract at 0.625 mg/mL, and the plate was reincubated for another 24 h. As an inflammation control, IL-1β (Invitrogen, Waltham, MA, USA) was used, while for basal activity, control plain medium was used. At the end of the assay, supernatants were collected, centrifuged to remove debris, and stored at −80 °C for further analysis.

Interleukins 6 (IL-6) and 8 (IL-8) and tumor necrosis factor alpha (TNF-α) were detected with an enzyme-linked immunosorbent assay (ELISA) using a Human IL-6 Elisa Kit High Sensitivity (Abcam, Cambridge, UK), a Legend Max Human Elisa Kit IL-8, and a Legend Max Human Elisa Kit TNF-α (BioLegend, San Diego, CA, USA), according to the manufacturer’s instructions. Interleukin values were obtained in picograms per milliliter of sample. To diminish the variability associated with any kind of proteomic-based assay, results are expressed in relative percentage of production to the interleukin levels in the basal (nonstimulated) control. The interleukin content of the basal control was set to 100%.

### 2.11. Statistical Analyses

Results are presented as the average ± standard deviation (n = 3). The normality of data distribution was tested by the Shapiro−Wilk test, and the null hypothesis that all means were equal was rejected when the difference between means was *p* < 0.05. Following the ANOVA, test multiple comparisons were performed for those statistically significant variables using Tukey’s post hoc test (homogeneity of variance was assumed at the *p* < 0.05 significance level). All statistical analyses were performed using STATISTICA version 14.0.0.15 (TIBCO, Palo Alto, CA, USA).

## 3. Results and Discussion

### 3.1. Sugarcane Straw Phenolic Compound Profile

From this analysis, we described the phenolic compounds detected in prepared sugarcane (*Saccharum officinarum* L.) straw extracts (50% (*v*:*v*) ethanol, 24 h, 1:10 *w*:*v* proportion at 30 °C) to maximize the bioactive compounds that confer higher bioactive properties.

The base peak chromatogram of a sample analyzed by LC-ESI-QqTOF-HRMS allowed for the full characterization of the phenolic compounds and the identification of three chemical classes of phenolic compounds, including hydroxybenzoic acids, hydroxycinnamic acids, and flavonoids, with 50 phenolic compounds identified (Figure 1 and Table 1). According to the obtained results, the developed method allowed us to detect and quantify the phenolic compounds; the determination coefficients obtained for each phenolic compound were ≥ 0.92 (Table 2). Thus, the obtained determination coefficients showed that the method was linear for the 18 phenolic compounds in the wide working ranges used (Table 2).

A similar phenolic profile was detected in sugarcane leaves of different genotypes [25], in sugarcane molasses extract [26], and sugarcane bagasse polyphenols extracted with 60% (*v*:*v*) ethanol assisted by ultrasound [27]. In all studies, the diversity of compounds identified was slightly different from those identified in this study; however, the groups of identified compounds were the same.

Within the hydroxybenzoic acids, eleven different compounds were identified; three of their most representative were 1-*O*-vanilloyl-β-D-glucose (306.55 ± 58.97 μg/g dw), 2,5-dihydroxybenzoic acid isomer 2 (301.78 ± 49.36 μg/g dw), and 4-hydroxybenzaldehyde (316.04 ± 40.13 μg/g dw), with *m/z* 329 [M−H]^−^ (C_14_H_17_O_9_), *m/z* 153 [M−H]^−^ (C_7_H_5_O_4_), and *m/z* 121 [M−H]^−^ (C_7_H_5_O_2_), respectively. The remaining compounds were present in quantities < 100 μg/g dw. This class of compounds represented 22% of all quantified compounds in the extract. Hydroxybenzoic acids derivatives are metabolites responsible for signaling in the defense response of plants to pathogens, and high levels are related to systemic infections [28]. In addition to plant functions, they have interesting biological properties, making them attractive as food supplements. 2,5-Dihydroxybenzoic acid revealed a moderate antidiabetic potential by interacting with receptors as DPP-IV and α-glucosidase [29]; 4-hydroxybenzaldehyde was reported to have a good capacity to inhibit *S. cerevisiae* biofilm formation [30].

The hydroxycinnamic acids were the most representative class, accounting almost half of the extract composition (43%). Fifteen compounds were identified, with the presence of quinic acid esterified with coumaroyl, caffeic, and ferulic acid units. Furthermore, we detected free ferulic acid, caffeic acid, and *p*-coumaric acid. Ferulic acid is an important molecular marker, because it contributes to the formation of lignin in mature cane, which varies among genotypes, cell types, and tissues in the same plant [31]. Caffeic, ferulic, and *p*-coumaric acids exhibited an immunomodulatory effect, which can be ascribed, in part, to their cytoprotective effect via their antioxidant capacity [32].

Chlorogenic acid (*m/z* 353 [M−H]^−^ (C_16_H_17_O_9_)) and 5-*O*-feruloylquinic acid (*m/z* 367 [M−H]^−^ (C_17_H_19_O_9_)) were present in high quantities: 407.28 ± 78.46 and 439.01 ± 97.24 μg/g dw, respectively. Caffeoylquinic acids derivatives were reported to have strong antimicrobial activity against microorganisms such as *Escherichia coli, Staphylococcus aureus, Enterococcus faecium, Proteus vulgaris, Pseudomonas aeruginosa, Klebsiella pneumoniae,* and *Candida albicans* [33].

Flavonoids are another important group of phenolic compounds widely represented in olive leaves. Among these, luteolin and apigenin, with moieties of glucose or arabinoside and *O*-glycosylated tricin, were detected as well as luteolin aglycone. The flavone C-glycosides, in addition to their physiological function in the plant, work as a defense mechanism against powdery mildew fungus [34]. The flavones identified in this work, mainly apigenin and luteolin, have been reported to have antioxidant activity [35,36]. Apigenin-8-C-glucoside isomer 2 (*m/z* 431 [M−H]^−^ (C_21_H_19_O_10_)), isoschaftoside (*m/z* 563 [M−H]^−^ (C_26_H_27_O_14_)), and tricin (*m/z* 329 [M−H]^−^ (C_17_H_13_O_7_)) presented higher contents among all the 24 flavones identified, with 288.09 ± 40.42, 257.76 ± 30.24, and 248.14 ± 3.40 μg/g dw, respectively. This class was also well represented in the extract, accounting for 40% of all identified phenolic compounds. In addition to a physiological function in the plant, the flavones, mainly apigenin and luteolin, have well-known antioxidant activity and therapeutic activity, such as antimalarial, antimicrobial, and antidiabetic effects [37]. According to several studies, tricin is suitable for clinical development due to its excellent pharmacological efficacy, bioactive properties, and low toxicity [38]. Tricin is considered a promising nutraceutical due to the properties that it exhibits, such as anticancer [39,40], antiobesity [41], antidiabetic [42], antioxidant [43,44], anti-inflammatory [45,46], and antiviral [47] activities.

### 3.2. Antioxidant Activity

The antioxidant capacity of the sugarcane straw extract powder was evaluated according two chemical methods (ABTS and DPPH), and the results are expressed in Trolox equivalents. According to the ABTS assay results, the extract possessed the capacity to neutralize free radicals at 53.1 mg TE/g; according to the DPPH assay results, the antioxidant capacity was 33 mg TE/g (Table 3). The difference observed between the methods may be explained by the richness of the compounds with the capacity to transfer electrons to a larger extent than protons (H^+^). Additionally, ABTS was performed under an aqueous environment, while DPPH was dissolved in methanol, resulting in the aprotic reaction environment being more adapted for determining the effects of less-polar compounds such as flavonoids [48].

The phenolic compounds present in sugarcane rid extracts were reported to possess potent antioxidant activity according DPPH and FRAP assays, and a strong correlation was observed with phenolic, flavonoids, and phytosterol contents [49]. A recent review summarized the antioxidant activity of different products and byproducts from sugarcane, where leaves and bagasse represent the parts of the plant with higher capacity to neutralize free radicals. This activity was related with the presence of polyphenols [4]. Another study suggested that sugarcane straw extracts possess antioxidant activity, which might be helpful in preventing or slowing the progress of various oxidative-stress-related diseases [50]. Current evidence suggests that the use of sugarcane juice as a natural source of dietary antioxidants in functional food is related to its highly abundant phenolics contents, especially flavonoids [51].

### 3.3. Antimicrobial Activity

The antimicrobial activity of sugarcane straw polyphenolic extracts is summarized in Figure 1. The extract did not show inhibitory or bactericidal activity against *E. coli*, *S. enterica*, *P. aeruginosa*, or *B. cereus*, although a reduction in growth occurred in *B. cereus*, especially in the highest concentration of extract tested (50 mg/mL). The extract showed a minimum inhibitory concentration (MIC) and minimum bactericidal concentration (MBC) against *S. aureus* at 50 mg/mL. *S. aureus* is an opportunistic pathogen, causing one of the most common foodborne diseases worldwide, which therefore has a negative impact related to the consumption of contaminated food and food products [52]. Even at the extract higher concentration tested (50 mg/ mL), we observed no bactericidal effect, but it still induced an inhibition of 74 ± 2.4% for *B. cereus*, 72 ± 2.7% for *S. aureus*, and 44 ± 14.2% for *S. enterica. E. coli* and *P. aeruginosa* showed no kind of growth inhibition once in contact with sugarcane extract (Figure 2). The results obtained in this study are in agreement with those in the literature, which described that Gram-positive bacteria are more susceptible to phenolic compounds than Gram-negative bacteria due to the different constituents and structures of cell membranes [53,54]. Phenolics are known to interfere with membrane proteins, inducing changes in their structure and function, which may affect electron transport, nutrient uptake, synthesis of proteins and nucleic acids, and enzyme activity [55]. Furthermore, some studies reported the antimicrobial activity of sugarcane-derived extracts. An extract from bagasse showed antimicrobial activity against pathogenic bacteria such as *Salmonella typhimurium* and *E. coli*, at an effective concentration of 2.50 mg/mL, while the inhibition of *S. aureus* was reported with 0.625 and 1.25 mg/mL for *Listeria monocytogenes* [54]. In addition, methanolic sugarcane root extracts had antimicrobial activity against *E. coli* and *Bacillus subtilis* with MICs of 0.276 and 0.271 mg/mL, respectively [56].

The natural antimicrobials market is expected to grow at an annual compound growth rate of 7.3% over the period 2019–2024, because multidrug-resistant food-borne pathogens have prompted the search for natural and cheaper antimicrobials that can replace the synthetic agents, representing a high economic impact [57].

Based on the results obtained in this study, the extract made from sugarcane straw showed potential to be used in combination with other novel preservation technologies to facilitate the replacement of traditional approaches as antimicrobials to control foodborne pathogens.

### 3.4. Effect on Tyrosinase Inhibition

Tyrosinase is multicopper enzyme responsible for enzymatic browning reactions in fruits and vegetables. Browning usually damages the color of plant-derived food products, which may indicate spoilage of their nutritional quality. This excessive tyrosinase activity in food can be prevented by using tyrosinase inhibitors [58]. Sugarcane phenolic extract (1–8 mg/mL) exhibited the capacity to inhibit tyrosinase between 28% and 73% in a dose-dependent manner (Figure 3). Kojic acid (0.021 mg/mL) was used as a positive control, which showed a capacity to inhibit tyrosinase of 88.50%.

Isolated compounds from sugarcane extract showed a good capacity to inhibit tyrosinase, and one structural characteristic from the most active compounds was the presence of the free hydroxyl group at position 4 in the aromatic ring, which is responsible for forming a hydrogen bond with the active site of the tyrosinase enzyme [59]. A commercial product (Officinol^TM^) developed for cosmetic application that is rich in polyphenols and is extracted from sugarcane molasses demonstrated the capacity to inhibit tyrosinase with an IC50 of 3.62 mg/mL [60].

Plant extracts rich in benzaldehyde and benzoate derivatives have a weak-to-moderate tyrosinase inhibitory activity; examples of these compounds include benzoic acid, benzaldehyde, anisic acid, anisaldehyde, cinnamic acid, and methoxycinnamic acid from the roots of *Pulsatilla cernua* [61]; 4-benzaldehydes from cumin [62]; *p*-coumaric acid from the leaves of *Panax ginseng* [63]; hydroxycinnamoyl derivatives from green coffee beans [64]; and vanillic acid and its derivatives from black rice bran [65]. Their inhibitory mechanism can be through Schiff bases with a primary amino group in the enzyme or by copper chelating mechanism [66,67]. A series of vanillin esters incorporating benzoic acid, cinnamic acid, and piperazine were reported to have tyrosinase inhibitory activity comparable to that of kojic acid [68]; *p*-hydroxybenzoic acid, chlorogenic acid, vanillic acid (4-hydroxy-3-methoxybenzoic acid), and protocatechuic acid from *Hypericum laricifolium* [69]; and ferulic acid from *Spiranthes sinensis* [70].

Enzymatic browning represents one of the food industry’s major problems, especially for fruits, vegetables, and seafood products. To prevent browning, food additives are used, including reducing agents and enzyme inhibitors. The food industry frequently uses ascorbic acid and various forms of sulfite-containing compounds as antibrowning agents. Several review articles have summarized the currently available tyrosinase inhibitors, which include polyphenols from synthetic, semisynthetic, and natural origins [67,71].

### 3.5. Antidiabetic Potential

α-Glucosidase is an oligosaccharide hydrolase located in the small intestinal epithelial cell brush border and is responsible for the hydrolysis of oligosaccharides and disaccharides into monosaccharides [72]. The inhibition of α-glucosidase activity potentially delays carbohydrate absorption and controls postprandial hyperglycemia, thereby helping to manage type 2 diabetes mellitus (T2DM) [73]. Several plant-derived polyphenols have shown strong inhibitory effects on α-glucosidase, which subsequently reduces the digestion rate of complex starches and oligo, tri-, and disaccharides into absorbable glucose [74].

The polyphenolic extract obtained from sugarcane straw was tested for α-glucosidase inhibition and compared with acarbose as positive control. Acarbose is the most used drug of this class, and the most widely studied; it was shown to increase life expectancy in patients with type 2 diabetes mellitus and reduce the risk of development of cardiovascular events in patients with impaired glucose tolerance [75]. The results showed that the polyphenolic extract tested had a α-glucosidase inhibitory effect between 15% and 38% at 1.25–5.00 mg/mL. Acarbose presented an α-glucosidase inhibition of 90.5% at 2.5 mg/mL (Figure 4). Although acarbose is commercially available and commonly administered to treat T2DM, several side effects, such as abdominal discomfort, flatulence and diarrhea, have been associated with its use [76].

Sugarcane molasses extract, rich in caffeic acid (11.64 mg/g), ferulic acid (10.49 mg/g), chlorogenic acid (1.77 mg/g), and gallic acid (0.87 mg/g), when used at 20 mg/mL, inhibited less than 30% of the α-glucosidase activity [77].

Among the most-cited polyphenols with proven α-glucosidase inhibition are *p*-coumaric acid and ferulic acid. This is most likely due to their strong binding capacity for the enzyme, as previously described in a study involving molecular docking analysis of a brown rice polyphenolic extract containing *p*-coumaric acid and ferulic acid [78]. Therefore, we expected significant α-glucosidase inhibition capacity from the tested sugarcane straw extracts because *p*-coumaric acid was one of the most abundant compounds (221.30 ± 31.28 µg/g dry extract). The potential mechanism of interaction between polyphenols and α-glucosidase reported the in literature describes the formation of hydrogen bonds or hydrophobic forces in the active site of the enzyme. For the case of ferulic acid, this mechanism increased the α-helix and decreased the β-sheet of α-glucosidase, preventing the substrate from binding to the enzyme [79]. The α-glucosidase inhibition was reported in the following order for free phenolic extracts: husk > bran > millet, with *p*-hydroxybenzaldehyde, vanillic acid, and *p*-coumaric acid being the predominant compounds, respectively. Husk extract had high contents of flavonoid C glycosides, with apigenin-C-pentosyl-C-hexoside and apigenin-C-dihexoside being the most representative flavonoids [80]. Similar C-glycosyl flavones to those reported in this study have shown the capacity to inhibit α-glucosidase. It was reported that 25 µM apigenin, vitexin, isovitexin, luteolin, orientin, and isoorientin inhibited α-glucosidase by 46.22, 19.91, 30.20, 60.42, 38.22 and 47.72%, respectively [81]. Apigenin inhibited the activity of α-glucosidase in a competitive manner, forming complexes where the main forces driving the interaction were hydrophobic interaction and hydrogen bonding [82].

Another way to improve glucose tolerance in diabetic patients is by enhancing the glucose-regulating hormones, namely glucagon-like peptide-1 (GLP-1) and glucose-dependent insulinotropic polypeptide (GIP) [83]. Dipeptidyl peptidase-IV (DPP-IV) is the enzyme responsible for the degradation of GLP-1, GIP [84], and gliptins, such as sitagliptin, vildagliptin, saxagliptin, alogliptin, linagliptin, gemigliptin, and teneligliptin, which are commercialized in the USA, Europe, Japan, and Korea as DPP-IV inhibitors [85]. However, natural alternatives that prolong the activity of GLP-1 and GIP and have no associated side effects are being investigated.

The polyphenolic extract from the tested sugarcane straw (i.e., 0.31–5.00 mg/mL) exhibited a strong capacity to inhibit DPP-IV (i.e., 62.27–99%). Sitagliptin, which was used as a positive control, inhibited DPP-IV activity by 96.76% when used at 0.041 mg/mL (Figure 5).

Chlorogenic acids isolated from aronia juice exhibited DPP-IV inhibitory activity, where 50% inhibition was achieved at 0.19 and 0.05 μM, respectively, for 3-caffeoylquinic acid and 4-caffeoylquinic acid. It was reported that the caffeoyl group formed hydrogen bonds with the active sites of the enzyme [86]. According to the literature, there are several chemical specificities described for flavonoids that enhance their capacity to inhibit DPP-IV: (a) glycosylation at the 3 position of ring A and 4′ position of ring B; (b) hydroxyl groups at the 5 and 7 positions of ring A and the 3′, 4′, and 5′ positions of ring B; (c) methoxylation at the 6 and 7 positions of ring A and the 3′ position of ring B; and (d) C2, C3-double bond in ring C, which promotes the interactions between the planarity, hydrogen bonding, and aromatic stacking [87].

One of the mechanisms proposed for polyphenols action on DPP-IV is a significant change in the natural conformation of the enzyme. DPP-IV is predominantly composed of α-helices, and polyphenols can turn them into β-sheet conformation or increase the random coil content, which leads to a partial unfolding of the polypeptide chain, disruption of the hydrogen bonding network structure, and rearrangement of hydrogen bonds, which may account for the reduced enzyme activity [88,89].

Although the obtained values of α-glucosidase and DPPIV inhibitions are inferior to those described for the commercial products currently in use as antidiabetic compounds, the results presented here clearly show the high potential of the tested extract. In addition, a natural extract must be obtained with green processes (i.e., using only one organic and food-grade solvent (ethanol), allowing the reintroduction of the extracted bioactive compounds into food products), within a circular economy framework (sugarcane byproduct), and potentially with considerably less (or even no) side effects, the tested polyphenolic extract may be a suitable and more sustainable alternative and used as a food supplement to help regulate T2DM.

### 3.6. Influence of Extract on Caco-2 Cell Viability

Before the Caco-2 inflammatory assay, a cytotoxicity assay was performed aiming to determine a safe and sublethal concentration of extract that could be used. In this study, Caco-2 cells were used because of their relevance to the analysis of the action of food components in intestinal epithelium issue [90].

The extract was dispersed in water at a concentration of 20 mg/mL, and the following concentrations were tested: 0.625–20 mg/mL. The lower concentration tested was the only considered not cytotoxic (0.625 mg/mL) after 24 h of treatment with the extract (Figure 6). The concentration of 1.25 mg/mL presented an inhibition of 60%, while concentrations above 2.50 mg/mL were considered toxic to the cells.

### 3.7. Anti-Inflammatory Effect

Inflammatory bowel disease is associated with dysregulation of the immune system, and the North America and European countries are the most affected. This disease induces an overexpression of interleukin 6 (IL-6) and tumor necrosis factor alpha (TNF-α) in the intestine [91]. The regulation of these biomarkers through diet may help with prevention and be used as coadjuvants for disease treatment [92].

The potential inflammatory-protective effect of sugarcane extract was evaluated using preincubation of the extracts in Caco-2 cell media and with coincubation (extract + IL-1β pro-inflammatory cytokine); the levels of three cytokines IL-6, IL-8 and TNF-α were evaluated (Figure 7). The extract was used at 0.625 mg/mL, because this concentration was the lowest concentration tested and considered physiological safe once in contact with Caco-2 cells. The inflammatory stimulus of Caco-2 cells was performed with IL-1β, which is a cytokine known to participate in the initiation and amplification of inflammatory activity leading to the production of biomarkers and inflammation mediators in Caco-2 cells [93].

In basal conditions, we detected the presence of IL-8 (9.6 pg/mL); in the presence of extract without stimulus, there was an increase in the production of TNF-α (27.99 pg/mL). After treatment with stimulus and with 0.625 mg/mL extract, no significant (*p* > 0.05) reduction in biomarker levels was detected compared with the positive control (challenge with IL-1β). In addition to not being statistically representative, we observed a tendency of decreased cytokines content, suggesting a potential anti-inflammatory pathway of the phenolic extracts. The TNF-α concentration reduced from 29.0 to 11.5 pg/mL, corresponding to a decrease of 60%, while that of IL-6 reduced from 35.0 to 23.2 pg/mL, and IL-8 had the lowest reduction (from 483.6 to 468.1 pg/mL) after stimulus with 10 µg/mL of IL-1β.

Most studies about the anti-inflammatory effect of polyphenols were conducted with extracts that contained a mixture of several compounds, and each class of phenolics can have a different effect on proinflammatory mediators. The mechanisms through which polyphenols exert their anti-inflammatory activity have not been well elucidated; however, it was hypothesized that they inhibit the synthesis of proinflammatory mediators through the modification of eicosanoid synthesis, inhibition of activated immune cells, or nitric oxide synthase and xyclooxygenase-2 via its inhibitory effects on NF-κβ [94,95].

Of the phenolic compounds, flavonoids from cocoa and tea have been found to modulate inflammation mediators such as IL-6 in a dose-dependent way. Those extracts were mainly rich in catechin and epicatechin derivatives [96,97]. It was found that apigenin exerted anti-inflammatory effects on human periodontal ligament cells by inhibiting LPS-induced production of NO, PGE2, IL-1β, TNF-α, IL-6, and IL-12 [98]. Protocatechuic acid was an effective anti-inflammatory agent, reducing nitric oxide production in microglial cells stimulated with lipopolysaccharide [99]. It was reported that ferulic acid had a strong anti-inflammatory effect, and it had a protective effect when the intestinal epithelial barrier dysfunction was promoted by LPS [100]. Chlorogenic and caffeic acids inhibited TNF-α and H_2_O_2_-induced IL-8 production in human intestinal Caco-2 cells [101]. Caffeoylquinic acid isomers and dicaffeoylquinic acids reduced IL-8 secretion in the Caco-2 human intestinal epithelial cell line [99].

Regarding sugarcane polyphenols, some studies described their potential as anti-inflammatory agents. A sugarcane stem aqueous extract (50 μmol gallic acid equivalent/L) induced a 1.5-fold decrease in IL-8 secretion when Caco-2 cells were treated with IL-1β [100]. A whole dried sugarcane ethanolic extract reduced NF-κB phosphorylation and IL-8 secretion in SW480 colon cancer cells stimulated with LPS [101].

## 4. Conclusions

Despite the numerous studies on the phenolic composition of sugarcane straw, this study highlights the potential of a derived byproduct extract as a multifunctional food ingredient. From a conventional and simple extraction made with 50% (*v*:*v*) ethanol, we recovered a high diversity of phenolic compounds belonging to three main classes, with hydroxycinnamic acids being most representative, followed by hydroxybenzoic acids and flavonoids. The extract exhibited potential as a food preservative, showing the capacity to neutralize free radicals as evaluated by two chemical methods (ABTS and DPPH). The extract also showed the capacity to inhibit tyrosinase, which is one of the enzymes responsible for food browning and inhibited the growth of three food spoilage bacterial species: *B. cereus > S. aureus* > *S. enterica.* The extract also exhibited nutraceutical potential because it inhibited the enzymes α-glucosidase and DPP-IV, indicating suitability as a food supplement for diabetes prevention. It also showed potential to help with controlling proinflammatory mediators (IL-6, IL-8, and TNF-α) when Caco-2 cells were stimulated with IL-1β. However, further biological assays should be employed to ensure effective and safe application in humans.

Thus, sugarcane straw could be considered a promising source of natural bioactive compounds that can be obtained within a sustainable and circular economy framework, with several applications as a food ingredient.

## Figures and Tables

**Figure 1 foods-11-04025-f001:**
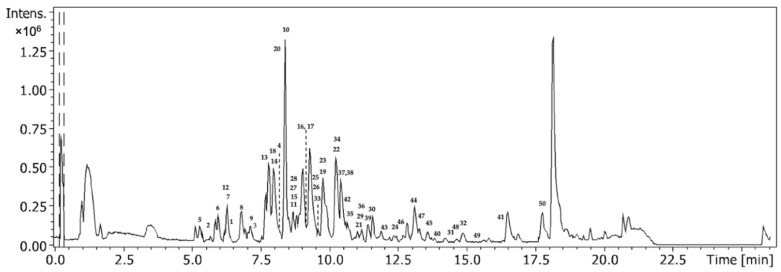
Base peak chromatogram (BPC) from sugarcane straw ethanolic extract (50% EtOH). Numbers 1-50 represent the identified compounds as described in Table 1.

**Figure 2 foods-11-04025-f002:**
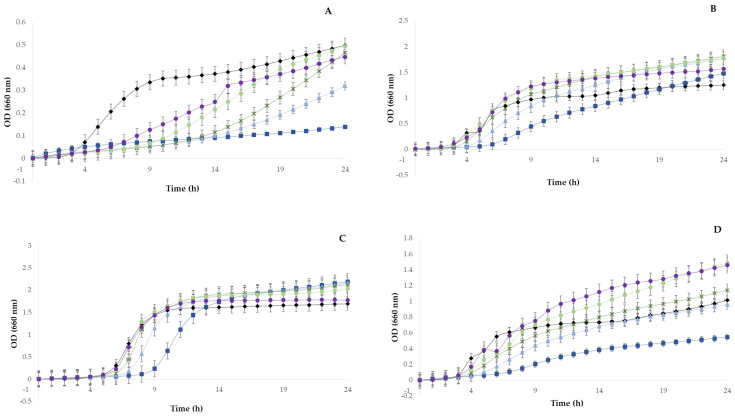
Optical densities (at 660 nm) (means ± SD) of *Staphylococcus aureus* (**A**), *E scherichia coli* (**B**), *Pseudomonas aeruginosa* (**C**), *Salmonella enterica* (**D**), and *Bacillus cereus* (**E**) when incubated with different concentrations of sugarcane straw polyphenolic extract: 10 mg/mL (○), 20 mg/mL (●), 30 mg/mL (×), 40 mg/mL (▲), 50 mg/mL (■), and positive control (◆). All determinations were carried out in duplicate, and results are shown as mean ± standard deviation.

**Figure 3 foods-11-04025-f003:**
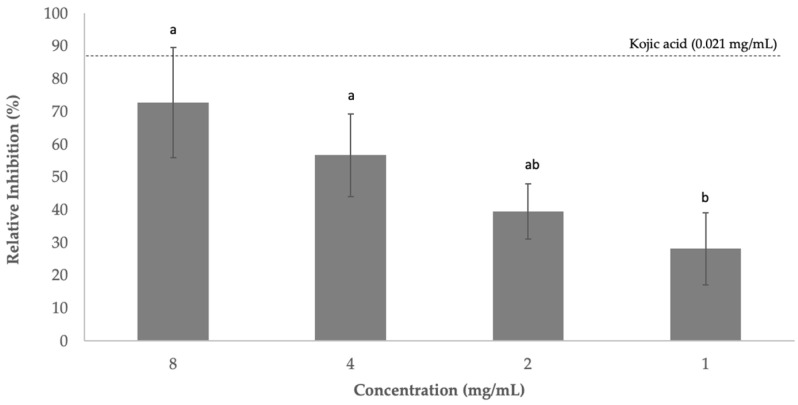
Percentage of tyrosinase activity inhibited (means ± SD) when tested with sugarcane straw extract. Extracts were previously diluted in water and tested at 8, 4, 2, and 1 mg/mL. Positive tyrosinase control used was kojic acid. Values presented are the mean ± standard deviation. ^a,b^ Different letters mean statistically significant differences were obtained (*p* < 0.05).

**Figure 4 foods-11-04025-f004:**
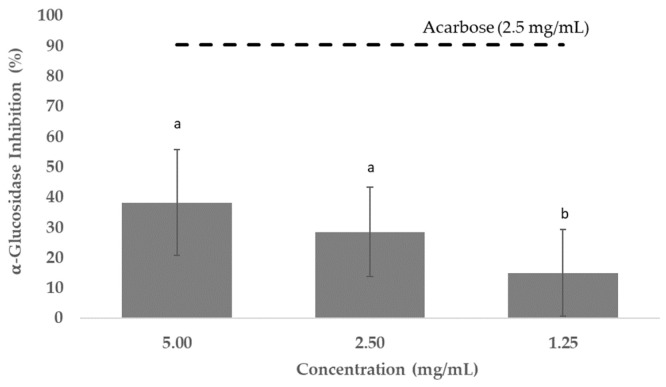
Percentage of α-glucosidase activity inhibited (means ± SD) tested with sugarcane straw extract. Acarbose (2.5 mg/mL) was used as positive control. ^a,b^ Different letters mean that statistically significant differences were obtained (*p* < 0.05).

**Figure 5 foods-11-04025-f005:**
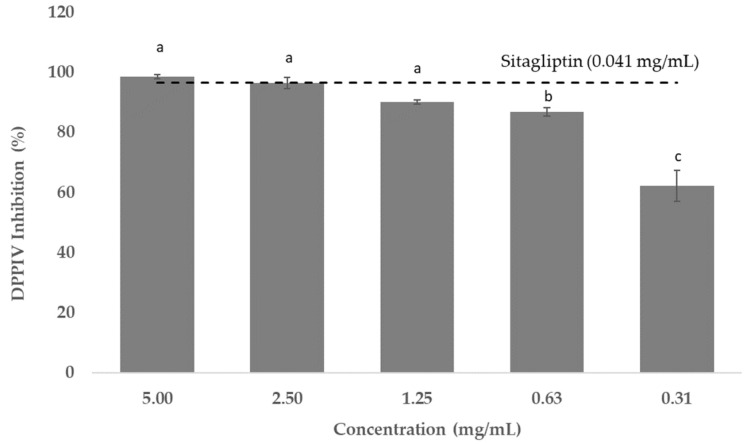
Percentage of DPP-IV activity inhibited (means ± SD) tested with the sugarcane straw extract. Sitagliptin (0.04 mg/mL) was used as positive control. ^a,b,c^ Different letters mean statistically significant differences were obtained (*p* < 0.05).

**Figure 6 foods-11-04025-f006:**
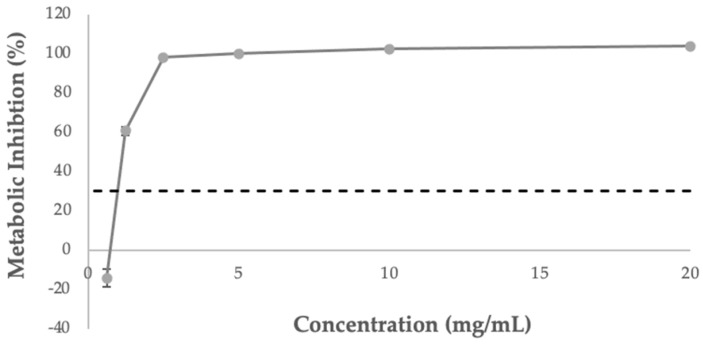
Cytotoxicity profile of sugarcane straw extract dispersed in water when in contact with intestinal Caco-2 cell line, measured by the potential of the extract to inhibit cell metabolic activity. The dotted line represents the 30% cytotoxicity limit, as defined by the ISO 10993-5:2009 standard (ISO 2009). Values are the mean ± standard deviation.

**Figure 7 foods-11-04025-f007:**
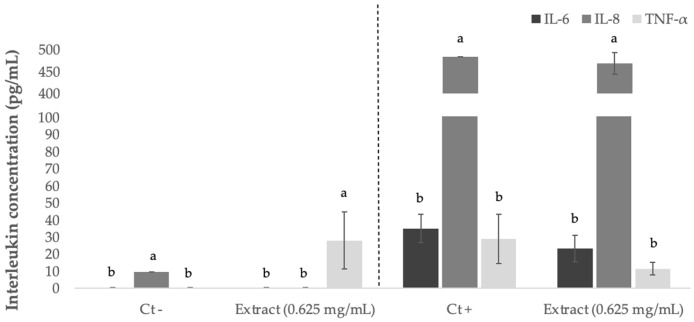
Interleukin-8 (IL-8), interleukin-6 (IL-6), and tumor necrosis factor (TNF-α) concentrations (mean ± SD) produced by Caco-2 cells when treated with sugarcane straw extract (0.625 mg/mL), using IL-1β as a proinflammatory stimulus. ^a,b^ Different letters mean statistically significant differences were obtained (*p* < 0.05).

**Table 1 foods-11-04025-t001:** LC-ESI-UHR-QqTOF-MS data of phenolic compounds identified and quantified (means ± SD, µg/g dry sugarcane straw extract).

	Proposed Name	Formula-H	*m*/*z* Theoretical Mass [M−H]^−^	*m*/*z* Measured Mass [M−H]^−^	Error (ppm)	MS/MS Fragments (*m*/*z*)	Concentration (µg/g Dry Extract)
	** *Hydroxybenzoic acids* **						
1	1-O-Vanilloyl-β-D-glucose	C_14_H_17_O_9_	329.0869	329.0878	2.6	167	306.55 ± 58.97
2	Protocatechuic acid	C_7_H_5_O_4_	153.0184	153.0193	3.3	109, 153	24.79 ± 3.70
3	2,5-Dihydroxybenzoic acid isomer 1	C_7_H_5_O_4_	153.0184	153.0193	3.2	109, 153	13.49 ± 1.34
4	2,5-Dihydroxybenzoic acid isomer 2	C_7_H_5_O_4_	153.0181	153.0193	3.3	65, 109	301.78 ± 49.36
5	Gentisic acid 2-O-β-glucoside	C_13_H_14_O_9_	315.0713	315.0722	2.7	108, 152	34.89 ± 9.66
6	Gentisic acid 5-O-β-glucoside	C_13_H_14_O_9_	315.0712	315.0722	2.9	109, 153	32.96 ± 6.97
7	Protocatechuic acid 4-β-glucoside	C_13_H_14_O_9_	315.0716	315.0722	1.6	109, 153	9.91 ± 2.32
8	4-Hydroxybenzoic acid	C_7_H_5_O_3_	137.0235	137.0244	4.6	137	21.99 ± 3.44
9	3,4-Dihydroxybenzaldehyde	C_7_H_5_O_3_	137.0233	137.0244	4.5	93, 137	53.69 ± 5.00
10	4-Hydroxybenzaldehyde	C_7_H_5_O_2_	121.0285	121.0295	3.0	121	316.04 ± 40.13
11	Syringic acid	C_9_H_9_O_5_	197.0445	197.0455	−1.9	123	97.21 ± 13.50
	** *Hydroxycinnamic acids* **						
12	Neochlorogenic acid	C_16_H_17_O_9_	353.0866	353.0878	3.4	135, 179, 191	239.43 ± 62.19
13	Chlorogenic acid	C_16_H_17_O_9_	353.0864	353.0878	4.0	191	407.28 ± 78.46
14	4-Caffeoylquinic acid isomer 1	C_16_H_17_O_9_	353.0866	353.0878	3.5	135, 173, 179, 191	200.28 ± 37.50
15	4-Caffeoylquinic acid isomer 2	C_16_H_17_O_9_	353.0864	353.0865	0.2	191	211.58 ± 19.16
16	cis-5-O-p-Coumaroylquinic acid isomer 1	C_16_H_17_O_8_	337.0919	337.0929	3.1	93, 163, 173, 191	27.46 ± 3.46
17	cis-5-O-p-Coumaroylquinic acid isomer 2	C_16_H_17_O_8_	337.0919	337.0929	3.1	191	12.41 ± 1.70
18	5-O-Feruloylquinic acid	C_17_H_19_O_9_	367.1021	367.1035	3.8	134, 193	439.01 ± 97.24
19	trans-3-Feruloylquinic acid	C_17_H_19_O_9_	367.1023	367.1030	3.5	173	247.79 ± 34.89
20	Caffeic acid	C_9_H_7_O_4_	179.0337	179.0350	−0.5	135, 179	106.33 ± 16.53
21	Ferulic acid	C_10_H_9_O_4_	193.0391	193.0506	4.3	134, 161, 193	60.28 ± 5.64
22	p-Coumaric acid	C_9_H_7_O_3_	163.0401	163.0401	−1.0	119	221.30 ± 31.28
23	Caffeoylquinic acid	C_16_H_17_O_9_	515.1220	515.1202	2.3	515	134.07 ± 15.60
24	4,5-Dicaffeoylquinic acid	C_25_H_23_O_12_	515.1196	515.1202	2.3	173, 179, 191, 335, 353	42.99 ± 1.77
25	Caffeoylshikimic acid isomer 1	C_16_H_15_O_8_	335.071	335.0772	−0.6	135, 161, 179	18.28 ± 1.43
26	Caffeoylshikimic acid isomer 2	C_16_H_15_O_8_	335.071	335.0772	−0.5	135, 161, 179	22.81 ± 0.86
	** *Flavonoids* **						
27	Apigenin-8-C-glucoside isomer 1	C_21_H_19_O_10_	431.1917	431.1923	1.4	89, 179	50.97 ± 7.67
28	Apigenin-8-C-glucoside isomer 2	C_21_H_19_O_10_	431.1912	431.1923	2.5	311, 341, 431	288.09 ± 40.42
29	Apigenin-8-C-glucoside isomer 3	C_21_H_19_O_10_	431.0982	431.1923	2.4	311, 341	37.18 ± 1.98
30	Apigenin-8-C-glucoside isomer 4	C_21_H_19_O_10_	431.0975	431.0984	2.1	311, 341	112.33 ± 4.81
31	Apigenin-8-C-glucoside isomer 5	C_21_H_19_O_10_	431.0984	431.0984	0.0	327, 341, 357	18.84 ± 1.75
32	Apigenin-8-C-glucoside isomer 6	C_21_H_19_O_10_	431.1352	431.0984	0.0	327, 357	88.45 ± 2.87
33	Isovitexin 2″-O-arabinoside	C_26_H_27_O_14_	563.1401	563.1406	1.9	353, 443	28.27 ± 1.68
34	Isoschaftoside	C_26_H_27_O_14_	563.1395	563.1406	1.9	353, 473	257.76 ± 30.24
35	Neoschaftoside	C_26_H_27_O_14_	563.1403	563.1406	0.5	399, 473	43.61 ± 3.82
36	Apigenin-6-C-arabinoside-8-C-glucoside	C_26_H_27_O_14_	563.1397	563.1379	−3.1	293, 413	63.86 ± 4.24
37	Luteolin-8-C-glucoside isomer 1	C_21_H_19_O_11_	447.0920	447.0933	2.9	327, 357	156.30 ± 8.42
38	Luteolin-8-C-glucoside isomer 2	C_21_H_19_O_11_	447.0920	447.0933	2.9	327, 357	87.15 ± 14.71
39	Vitexin 2″-O-beta-L-rhamnoside	C_27_H_29_O_14_	577.1559	577.1563	0.7	293, 413	62.21 ± 0.86
40	Apigenin 7-O-neohesperidoside	C_27_H_29_O_14_	577.1556	577.1563	1.2	293, 413, 473	59.86 ± 0.86
41	Luteolin	C_15_H_9_O_6_	285.0407	285.0405	−1.0	285	66.56 ± 1.30
42	6-Methoxyluteolin 7-rhamnoside isomer 1	C_22_H_21_O_11_	461.1083	461.1089	1.4	461	42.85 ± 4.38
43	6-Methoxyluteolin 7-rhamnoside isomer 2	C_22_H_21_O_11_	461.1088	461.1136	2.3	341, 371	36.61 ± 0.98
44	Tricin-O-neohesperoside isomer 1	C_29_H_33_O_16_	637.1772	637.1774	−3.9	329	60.07 ± 0.11
45	Tricin-O-neohesperoside isomer 2	C_29_H_33_O_16_	637.1775	637.1638	−0.1	329	53.80 ± 0.64
46	Tricin-7-O-glucoside	C_25_H_31_O_10_	491.1919	491.1823	0.7	329	132.62 ± 7.58
47	Tricin-7-O-rhamnosyl-glucuronide	C_29_H_31_O_17_	651.1570	651.1567	−0.4	329	76.29 ± 1.34
48	Tricin-4-(O-erythro) ether glucoside isomer 1	C_33_H_35_O_16_	687.1941	687.1786	3.1	165, 195, 329, 491, 525	84.31 ± 3.25
49	Tricin-4-(O-erythro) ether glucoside isomer 2	C_33_H_35_O_16_	687.1937	687.1786	3.0	165, 195, 329, 491, 526	69.40 ± 4.25
50	Tricin	C_17_H_13_O_7_	329.0664	329.0667	1.0	299	248.14 ± 3.40

**Table 2 foods-11-04025-t002:** Parameters of calibration curves used for sugarcane straw phenolic compound quantification through LC-ESI-UHR-QqTOF-MS.

Standard	Concentration Range (µg/mL)	Equation Curve	Determination Coeficiente (R^2^)	LOD (µg/mL)	LOQ (µg/mL)
Vitexin	0.06–1.96	*y* = 10889817 *x* + 98542	0.96	0.23	0.71
Diosmetin	0.06–1.90	*y* = 3875491 *x* + 774787	0.96	0.66	2.00
Isoschaftoside	0.02–0.70	*y* = 9619455 *x* + 380002	0.98	0.17	0.53
Orientin	0.03–0.95	*y* = 3245142 *x* + 223820	0.98	0.28	0.84
Vitexin-2-*O*-rhamnoside	0.06–1.96	*y* = 3617818 *x* + 550052	0.99	0.28	0.84
Tricin	0.01–0.47	*y* = 8999089 *x* + 145571	0.98	0.14	0.42
Luteolin	0.06–1.96	*y* = 6729048 *x* + 1778703	0.96	0.74	2.25
Protocatechuic acid	0.09–1.40	*y* = 1586510 *x* + 60821	0.99	0.29	0.72
Vanillic acid	0.02–0.64	*y* = 1045485 *x −* 46131	0.99	0.24	0.72
*p*-Coumaric acid	0.14–1.14	*y* = 3125587 *x* + 40037	0.98	0.44	1.33
Caffeic acid	0.15–1.24	*y* = 2508428 *x* + 118063	0.95	0.80	2.42
Ferulic acid	0.13–1.07	*y* = 1143919 *x* + 25137	0.98	0.48	1.45
Chlorogenic acid	0.12–0.95	*y* = 779052 *x* + 52629	0.92	0.85	2.59
2,5-Dihydroxybenzoic acid	0.15–1.18	*y* = 1852876 *x* + 85535	0.99	0.40	1.21
4-Hydroxybenzaldehyde	0.15–1.19	*y* = 8654434 *x −* 50043	0.99	0.37	1.13
4-Hydroxybenzoic acid	0.14–1.12	*y* = 5072196 *x* − 20201	0.99	0.11	0.34
3,4-Dihydroxybenzaldehyde	0.14–1.20	*y* = 2190795 *x* + 65616	0.99	0.35	1.07
Syringic acid	0.16–1.26	*y* = 405197 *x* + 14954	0.93	1.15	3.49

**Table 3 foods-11-04025-t003:** Antioxidant activity (means ± SD) of polyphenolic sugarcane straw extract measured through ABTS and DPPH.

Antioxidant Activity	ABTS	DPPH
	(mg TE/g Dry Extract)	
Sugarcane straw extract	53.1 ± 0.0	33.0 ± 0.0

## Data Availability

This study did not report any data.

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
