# Peer review of "Sugarcane Straw Polyphenols as Potential Food and Nutraceutical Ingredient"

_foods, 2022, doi:10.3390/foods11244025_

Round 1

Reviewer 1 Report

Title: Sugarcane straw polyphenols as potential food and nutraceutical ingredient

Comments:

Line no 24: Which antioxidants are mainly present in straw?

Line no 25: How the intestinal inflammation will be reduced?

Line no 26: Abstract is very well written and comprehensive.

Line no 31: Add this latest paper here

Potential impact of ultrasound, pulsed electric field, high-pressure processing and microfludization against thermal treatments preservation regarding sugarcane juice (Saccharum officinarum)

Line no 36: Mention bioactive compounds.

Line no 49: What type of quality deteriorations?

Line no 54: Add this paper

Combined effect of microwave and ultrasonication treatments on the quality and stability of sugarcane juice during cold storage

Line no 63 to 65: No need to add this information.

Line no 67: Naturally derived food-derived molecules, give examples.

Line no 94: Appreciated theme and it would be a good contribution towards food industries.

Line no 332: Tables data is very appreciable.

Line no 371: What is the main reason for this neutralization?

Line no 414: Graphical work is done very well.

The conclusion is very short and needs to be improved and more comprehensive.

Author Response

First, at all, we would like to thank all the suggestions given by the reviewers under supervision of the editor, author’s have carefully undertaken the revision requests for your reconsideration. All the changes made in the manuscript are highlighted at blue color directly in text and the identified in the reviewer’s letter with the line number.

Title: Sugarcane straw polyphenols as potential food and nutraceutical ingredient

Comments:

Line no 24: Which antioxidants are mainly present in straw?

The main phenolic compounds present in sugarcane straw are mentioned on line 19, and the phenolics compounds are the ones responsible for the antioxidant capacity of the extract. This information was highlighted on line 29.

Line no 25: How the intestinal inflammation will be reduced?

Intestinal inflammation could be controlled trough the reduction of the inflammatory mediators such as IL-6, IL-8 and TNF-alpha. In line 32 can be read “…co-adjuvant to reduce intestinal inflammation by reducing the pro-inflammatory mediators.”

Line no 26: Abstract is very well written and comprehensive.

Author’s thanks for the commentary.

Line no 31: Add this latest paper here

Potential impact of ultrasound, pulsed electric field, high-pressure processing and microfludization against thermal treatments preservation regarding sugarcane juice (Saccharum officinarum)

Reference added now on line 38.

Line no 36: Mention bioactive compounds.

The term bioactive compounds were referring to phenolic compounds in general without specifying anyone. The term was changed now on line 42.

Line no 49: What type of quality deteriorations?

Sentence was corrected, since it was mentioning to the deterioration of lipid components, now on line 55.

Line no 54: Add this paper

Combined effect of microwave and ultrasonication treatments on the quality and stability of sugarcane juice during cold storage

Reference added now on line 60.

Line no 63 to 65: No need to add this information.

The purpose of this sentence was to highlight the importance that alternative antidiabetic ingredients may have since the diabetes type 2 is rising around the world. Because of that author’s still consider this information relevant so it was decided to not delete this information from the introduction.

Line no 67: Naturally derived food-derived molecules, give examples.

Examples were given on lines 74-75: “Thus, naturally food-derived molecules such as flavonoids, alkaloids, terpenoids, anthocyanins, glycosides, phenolic compounds…”.

Line no 94: Appreciated theme and it would be a good contribution towards food industries.

Authors did not understand the reviewer intention with this comment.

Line no 332: Tables data is very appreciable.

Authors appreciate the comments. The table information was improved.  

Line no 371: What is the main reason for this neutralization?

The main reason for those properties is related with richness in phytochemicals like phenolic compounds. Sentence was rewritten now on line 406.

Line no 414: Graphical work is done very well.

Authors appreciate the comments.

The conclusion is very short and needs to be improved and more comprehensive.

Conclusion should not be very descriptive, for that reason authors try to write a short summary with the main conclusions of the work. A sentence was added to conclusions to demonstrate the application of the extract may have due to their described properties in this work.

Reviewer 2 Report

I reviewed the manuscript entitled, Sugarcane straw polyphenols as potential food and nutraceutical ingredients. The manuscript is well written and contributes to the field. Authors performed the complete study from phenolic analysis to antidiabetic activity including cytotoxicity. This study further contributes to the concept of waste valorization and its possible application in food and nutraceuticals.
addressing below suggestions:

Abstract: background of the study and need of conducting this study should be introduced

Please provide ref for section 2.3. Extraction and isolation of phenolic compounds from sugarcane straw

Please provide the ref for section 2.4. Phenolic compounds and organic acids identification and quantification by LC-ESI-UHR- QqTOF-MS

Please provide ref for section 2.7. Inhibition of tyrosinase activity quantification

Provide the ref for the section 2.8.2. Dipeptidyl peptidase-IV (DPP-IV) inhibition assay and 2.9

Line 300 358, 391 431 480 523 561: what is (Error! Reference source not found.).??

Lines 378-381….. something is missing

Results and discussion are appropriate

References are not according to the journal format. Please revise it

Author Response

First, at all, we would like to thank all the suggestions given by the reviewers under supervision of the editor, author’s have carefully undertaken the revision requests for your reconsideration. All the changes made in the manuscript are highlighted at blue color directly in text and the identified in the reviewer’s letter with the line number.

I reviewed the manuscript entitled, Sugarcane straw polyphenols as potential food and nutraceutical ingredients. The manuscript is well written and contributes to the field. Authors performed the complete study from phenolic analysis to antidiabetic activity including cytotoxicity. This study further contributes to the concept of waste valorization and its possible application in food and nutraceuticals.

addressing below suggestions:

Abstract: background of the study and need of conducting this study should be introduced.

A sentence was added in the abstract on lines 11-13 describing that sugarcane straw represent a high-volume waste with environmental impact, and with this there’s being wasting a valuable source of polyphenols that may have interest for food as a new ingredient, creating more value to the industry.

Please provide ref for section 2.3. Extraction and isolation of phenolic compounds from sugarcane straw

The extractions procedure was internally optimized in our laboratories but not published. Because the focus of this manuscript was the biological properties of sugarcane straw extract was to demonstrate its potential application in food industry no more information regarding extraction procedure was mentioned.

Please provide the ref for section 2.4. Phenolic compounds and organic acids identification and quantification by LC-ESI-UHR- QqTOF-MS

Reference was added on line 146.

Please provide ref for section 2.7. Inhibition of tyrosinase activity quantification

This method was performed with a commercial kit of tyrosinase inhibitor screening kit (ab204715, abcam) according to the methodology described by the supplier as described on lines 216-217. Because of that no bibliographic reference was indicated for this method.

Provide the ref for the section 2.8.2. Dipeptidyl peptidase-IV (DPP-IV) inhibition assay and 2.9

Section 2.8.2 was performed with a commercial kit of DPP (IV) inhibitor screening assay kit and the authors followed the procedure described by the supplier (Cayman Chemical, USA) as can been read on lines 252.

Section 2.9 had references [21, 22] as mentioned on line 270.

  1. Costa, E.M.; Pereira, C.F.; Ribeiro, A.A.; Casanova, F.; Freixo, R.; Pintado, M.; Ramos, O.L. Characterization and Evaluation of Commercial Carboxymethyl Cellulose Potential as an Active Ingredient for Cosmetics. Appl. Sci. 2022, 12, doi:10.3390/app12136560.
  2. Machado, M.; Costa, E.M.; Silva, S.; Rodriguez-Alcalá, L.M.; Gomes, A.M.; Pintado, M. Pomegranate Oil’s Potential as an Anti-Obesity Ingredient. Molecules 2022, 27, doi:10.3390/molecules27154958.

Line 300 358, 391 431 480 523 561: what is (Error! Reference source not found.).??

During the submission of the word file probably something went wrong, and the figures and tables references appears with this error. The references to figures and tables were corrected and added to the manuscript highlighted with blue color.

Lines 378-381….. something is missing

Line 420: can be read “Figure 2” that was the missing information, now added.

Results and discussion are appropriate

References are not according to the journal format. Please revise it

References were revised according to the journal format, and for that it was used the Mendeley software.

Reviewer 3 Report

Very interesting article, very well written. Congratulations to the Authors of this work.

All comments in the file.

Author Response

First, at all, we would like to thank all the suggestions given by the reviewers under supervision of the editor, author’s have carefully undertaken the revision requests for your reconsideration. All the changes made in the manuscript are highlighted at blue color directly in text and the identified in the reviewer’s letter with the line number.

Please expand the abbreviations used in the abstract.

As suggested by the reviewer all the abbreviations in the abstract were expanded and were highlighted in blue color.

I understand that this is a research hypothesis. Please refer to it in the conclusions.

Because it was a hypothesis authors decided to delete the sentence from introduction and transfer to conclusion section.

The symbol of the oxygen atom in the names of phenolic compounds should be written in italics. Please use throughout the manuscript.

The symbol was corrected in all manuscript.

Please keep the following record for each equipment used throughout the manuscript: name (model, manufacturer, country, city).

The equipment list was verified and corrected according to reviewer comments.

I am asking for information about technological and apparatus repetitions.

Author’s did not understand this comment. We would like reviewer to specify what is intended with this comment made on pag 3 line 118.

w/v - italics;

please add such an entry for each percentage concentration

w/v was corrected to italics and added to each percentage concentration as suggested by the reviewer.

How exactly was the quantification carried out? Please add additional information about the calibration of the device in the materials: for which compounds the standardization was carried out, in which concentration range, please provide the equation of the curve, its fitting, LOD, LOQ, linearity range.

As suggested by the reviewer, a table with calibration curves information (equation curve, linearity, and concentration range, LOD and LOQ) was added to the results and discussion section and the standards used as the information about external calibration added to methods section.

Please use the following wording everywhere: mg/mL - pay attention to accidental and double spaces

Text was verified.

For each kit used - please provide the exact catalog number.

The catalog number for DPP-IV kits was added on line 252. Tyrosinase kit already had catalog number as ab 204715 on line 217.

Please standardize the notation of all methodologies, if any abbreviation, type of provision etc. is used once. - it's the same in every paragraph.

Whole manuscript revised.

please correct (Error! Reference source not found)

During the submission of the word file probably something went wrong, and the figures and tables references appears with this error. The references to figures and tables were corrected and added to the manuscript highlighted with blue color.

m/z – italics

m/z changed to italics.

Please add a representative chromatogram from the analysis

A figure from the base peak chromatogram (BPC) was added to manuscript as Figure 1.

Please give theoretical and real mass errors - in ppm

As suggested by the reviewer the theoretical and real mass were added to the table as well error in ppm.

if a given value is given as a decimal approximation, i.e. 53.1 - then the standard error should be written in the same format, i.e. 0.0 use throughout the manuscript

Values were corrected, for uniformization.

please use colors

Antimicrobial graphics were changed to colors.

Specifying values above 100% for inhibition is, in my opinion, incorrect DPP-IV

We would like to thanks’ the reviewer to highlight this. The raw data was verified and there was a calculation error, and the graphic was corrected for values below 100 % inhibition.

Round 2

Reviewer 3 Report

Accept in present form.